# Diatom Voucher Flora and Comparison of Collection and Taxonomic Methods for Biodiversity Hotspot Upper Three Runs Creek

**Katherine M. Johnson** [1,2], **Evelyn Gaiser** [2] and **Kalina M. Manoylov** [1,*]

1   Department of Biological and Environmental Sciences, Georgia College & State University, Milledgeville, GA 31061, USA; kajohnso@fiu.edu
2   Institute of Environment and Department of Biological Sciences, Florida International University, Miami, FL 33199, USA; gaisere@fiu.edu
*   Correspondence: kalina.manoylov@gcsu.edu

**Abstract:** Incorporating diatoms species and their autecology from reference stream conditions is essential for improving the accuracy of North American diatom bioassessments. This study documents a voucher flora and physicochemical conditions of Upper Three Runs Creek (UTRC), a tributary to the Savannah River that has been protected from heavy human activity for the last 50 years. The algae of UTRC and its watershed have been monitored continuously during this time by the Academy of Natural Sciences in Philadelphia to detect potential impacts from the Savannah River Site and Plant Vogtle. Standard protocols were used to sample and denote substrate types and preferences and to estimate relative abundances of diatom species. Data from artificial substrates (diatometers) were compared to composite samples. Phenotypic plasticity of taxa from *Gomphonema parvulum*, *Eunotia incisa*, and *Tabellaria flocculosa sensu lato* species complexes were considered in biodiversity metrics. We provide documentation of these separations. A total of 297 species/operational taxonomic units (OTUs) were recorded. For 2018 samples, *Eunotia rhomboidea* accounted for 8.3% for separated taxa methods and *G. parvulum sensu lato* (16.2%) dominated for combined taxa methods. *Luticola goeppertiana* (5.4%) was the most abundant taxon in the 1956 samples. The 1956 composite samples species richness means were greater and significantly greater than those for other sample types (ANOVA, df = 3, $p = 0.004$). The recounted 1956 composite samples had the highest species richness (153) followed by the left diatometer (129). At the same location, the right and left diatometers were not similar. Both methods (diatometers and composite sampling) are useful when assessing species richness. Diatom community composition indicated an acidic and highly oxygenated environment.

**Keywords:** Upper Three Runs Creek; diatoms; voucher flora; southeastern biodiversity; Savannah River; diatometers; algal sampling; taxonomic methods; aquatic bioindicators



## 1. Introduction

Diatoms have long since been used for biological assessments of water quality and aquatic ecosystem health [1,2], but the reliability of these assessments depends on careful characterization of the species and their environmental preferences along a narrow environmental gradient of reference/undisturbed conditions [3]. Because algal species can be protected under U.S. environmental policy (Endangered Species Act of 1973, 16 U.S.C. § 1531 et seq), creating diatom voucher flora for reference or control site streams aids in the protection of the freshwater resources in which they are found. Successful application of diatoms as environmental indicators also requires consistent identification [4] and repeatable practices [5]. For instance, a 2008 study investigated traits and species composition of benthic diatoms to evaluate the accuracy of regional and western U.S. indicators of stressors of environmental conditions [6]. The authors concluded that because water

chemistry and land use varied greatly in streams, incorporating reference conditions of a narrower environmental gradient could develop a more accurate assessment. There is a particular paucity of studies documenting the diatom flora of the acidic, blackwater rivers of the southeastern United States despite activities to monitor these waters for effects of pollution. This study documents the diatom flora and environmental conditions of Upper Three Runs Creek (UTRC), a tributary and headwater stream of the Savannah River and is known as a southeastern biodiversity hotspot [7].

We chose UTRC as a study site for the biodiversity it comprises, and the reference conditions it provides for the Savannah River. Located along the Savannah River, in South Carolina, is the U.S. Department of Energy's (DOE) Savannah River Site (SRS). The research site in the UTRC is located within a protected area of the SRS complex, which is designated by the SRS to receive minimal land use impacts and serve as a control site in scientific studies. With a drainage area of 25,900 km$^2$ (10,000 square miles), which drains over the states of Georgia, South Carolina, and North Carolina, the Savannah River is one of the major rivers in the southeast providing potable water to an estimated 1.5 million people making the Savannah River a valuable surface water resource [8]. Past studies of UTRC and the upper reaches of the Savannah River have used diatoms to assess ecosystem health; however, they have not provided a clear voucher flora for UTRC as a reference site.

We incorporated physicochemical data into analyses for this study, to provide site conditions for diatom species found in our UTRC voucher flora. For this study, we recorded parameters, such as pH, temperature, dissolved oxygen, and conductivity values. We did not conduct nutrient analysis at this UTRC site. However, taxonomic identifications coupled with the autecology literature provided probable inferences about the type of nutrient environment found at this site.

*Past Monitoring of Upper Three Runs Biodiversity*

From the 1950s to early 2000s the Academy of Natural Sciences of Philadelphia (ANSP) conducted water quality surveys along the Savannah River using diatoms as biological indicators [9,10]. During the Savannah River surveys, ANSP also collected biological samples from Upper Three Runs Creek (UTRC). The ANSP conducted these surveys for Westinghouse Savannah River Company to monitor and evaluate potential impacts from SRS and Plant Vogtle [10]. The SRS was originally constructed in the 1950s to produce plutonium and tritium for nuclear weapons during the Cold War [11]. Today, the site is responsible for environmental cleanup, disposing of nuclear waste, and developing energy and defense technologies. The SRS covers an area of approximately 777 km$^2$ (300 square miles), which encompasses three counties: Aiken, Allendale, and Barnwell. In 1972, the SRS was designated as a National Environmental Research Park encompassing a variety of habitats (wetlands, hardwood stands, and riparian ecosystems).

Little remains known about southeastern U. S. diatom species. What is known about southeastern U. S. diatom species has been largely contributed by late phycological pioneer Dr. Ruth Patrick in these ANSP Savannah River reports and detailed in her monographs from 1966 [12,13]. These ANSP reports and archives are important because they provide long term ecological monitoring data of the Savannah River and southeastern United States diatom species. In the 1950s, the ANSP started these surveys with composite sampling and later switched to using diatometers as a collection method. In a synthesis of data given in the ANSP 2000 report (using diatometers), upstream sites were shown to have higher water quality when algal biodiversity was used as a proxy and algal biodiversity was also shown to decrease over time regardless of seasonality [10]. Physicochemical data and voucher flora for taxonomic identification were not provided in these biological assessment reports. In this study, we compare composite and diatometer sampling methods and investigate biodiversity from our site and an ANSP 1956 UTRC site.

The majority of past phycological work has been conducted in Europe, and American diatomists have used European names and metrics to infer regional and national water quality. This has led to assigning European published names to southeastern diatom

species, which could change perspectives on freshwater conditions in these areas. Because algal research conducted throughout Europe remains a primary literature source for diatom identification, documentation of North American diatom species and their autecology is imperative [14–16]. A 2007 study compared North American lists to European taxa assignments and found that metrics created for U.S. rivers were better than those created for Europe when assessing water quality [17]. Results from their study also found that combining regional and national metrics was better at assessing water quality than using only national criteria. The authors demonstrate the importance of using local metrics and agreement of taxonomic identification for aquatic assessments. Due to the need of continuous research on U.S. southeastern diatoms, European literary resources as well as American were used in our study. Together with a recent southeastern diatom voucher flora [18], our voucher flora provides a reference to standardize local and regional diatom species identification for future long term ecological monitoring of UTRC and the Savannah River.

A 2012 study along UTRC found high morphological variability among representatives of known species complexes: *Gomphonema parvulum sensu lato*, *Eunotia incisa sensu lato*, and *Tabellaria flocculosa sensu lato* [19]. In this study, we investigate a site along UTRC from the taxonomic perspective. We chose this perspective to standardize local species identification for long-term monitoring of UTRC, to better understand species richness at a known reference site with high biodiversity, and to contribute to the North American literature on diatom species. Due to the taxonomic nature of this study, in an effort to identify all valves (cells) to species level, we only sampled one area along UTRC. We do not make assumptions for the creek as a whole in our findings. To evaluate past biodiversity and morphological variability of species from UTRC, we used archived material from 1956 composite samples and laboratory resources located in the ANSP Diatom Herbarium. However, we conducted all analysis and algal enumeration from these archives. Analysis and enumeration were not conducted by the ANSP. All images for the voucher flora for our site are from samples collected for this study in 2018. Our objectives include: (1) assessing biodiversity at a site along UTRC and creating a voucher flora; (2) determining ecological indication from the diatom species collected at this site as well as physicochemical parameters, such as pH, temperature, dissolved oxygen, and conductivity; (3) comparing sampling methods (diatometers vs. composite samples) using biodiversity and similarity metrics.

## 2. Materials and Methods

### 2.1. Study Site

Field observations were conducted at UTRC, located at (Lat. 33.393067, Long. −81.610719). This study site was located upstream of a previous 2012 GCSU study site (Lat. 33.370750, Long. −81.627738) due to bridge construction. This site is located between a range of 50 m and 155 m above sea levels (Figure 1). UTRC has a 255-km$^2$ basin and is a tributary, which drains into the Savannah River. Georgia and South Carolina border the Savannah River, which is 483 km long and comprises a basin of 25,900 km$^2$. This site is also designated by SRS to receive as minimal land use impacts as possible. Located in a protected area and designated as a biodiversity hotspot, UTRC provides a unique habitat for studying diatom taxa [7].

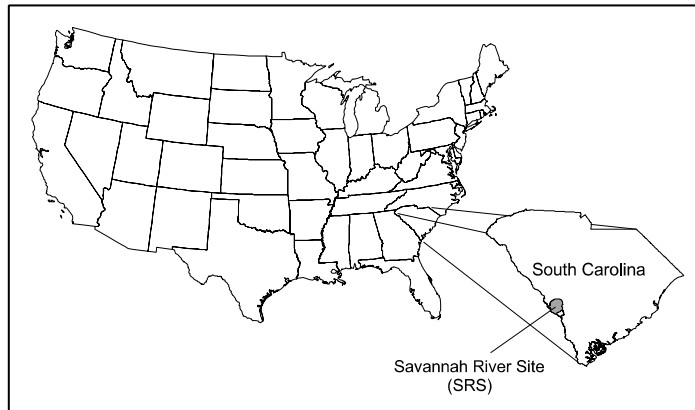

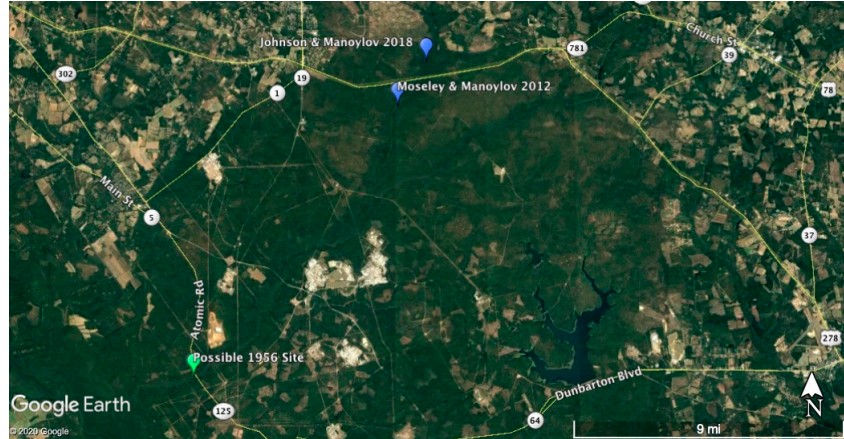

**Figure 1.** Map (**top**) of the United States depicting the location of the SRS from 2003 Savannah River Site Environmental Report www.srs.gov, accessed on 16 April 2020. Map (**bottom**) showing the sampling location of past GCSU studies (Lat. 33.370750, Long. −81.627738) and this 2018 study (Lat. 33.393067, Long. −81.610719) on UTRC, which is located on SRS, ©2020 Google.

## 2.2. Sampling

Diatoms were collected with diatometers, a plastic housing device attached to floats, which hold a total of seven glass slides. Diatometers were deployed on both the right and left sides of the creek for a period of 18 days from 23 March to 9 April 2018, following methodologies specified in past ANSP reports. Because of the differences in widths of the Savannah River and UTRC, only two diatometers were used. However, triplicate samples were taken from each diatometer for analysis, and the remaining four slides were archived in the GCSU Manoylov Phycology Lab. These devices are commonly used in algal monitoring studies to standardize the substrate area of algal colonization. Because these devices select for adnate species, triplicate composite samples were also taken at this site at the time of diatometer retrieval. Sampling followed the 2005 APHA standard methods for examination of water and wastewater and currently used 1999 EPA periphyton protocols [20,21]. Composite sampling consisted of collecting triplicate samples of at least 250 mL of a representative sample of site vegetation, woody debris, silt, and sand. Half of each triplicate was processed with nitric acid, and 125 mL were preserved and archived in the GCSU Manoylov Phycology Lab. Physicochemical data were collected at the time of diatometer deployment and retrieval. These data were collected with a YSI 556 MPS (multi-probe system) that measured pH, temperature (°C), dissolved oxygen (%, mg/L), and conductivity (mS/cm$^2$) simultaneously. Measurements were recorded once the instrument readings stabilized.

*2.3. Laboratory Methods*

2.3.1. Processing

After collection, archived composite samples were preserved with a 3% final concentration of formaldehyde. Before preservation, 30 mL of 70% nitric acid was added to samples for heat and potassium dichromate digestion. Samples were digested for one day and then decanted daily for eight days with deionized water or until sample yielded a pH of 7. After decanting, excess water was siphoned and samples were concentrated in 20 mL vials. From these vials, an optimum amount of suspension from each slide was mixed with at least 1 mL of deionized water and distributed on 22 × 22 mm coverslips to dry. After drying, Naphrax® (Brunel Microscopes Ltd., Chippenham, Wiltshire, UK) mounting medium was used to make permanent slides. These methods follow those detailed in Moseley and Manoylov [19]. Diatometer samples were prepared in a similar manner, except all processing was conducted within a 50 mL falcon tube so as not to lose frustules from scraping slides before nitric acid digestion. Slides from diatometers were selected using Microsoft Excel 2010 random generator. After being selected, slides were placed in tubes. Twenty-seven to 30 mL of hydrogen peroxide 30% concentration were added to the tubes. Tubes were then placed in a 200 °C water bath for 2–3 h before flipping slides within tubes to ensure entire immersion into the hydrogen peroxide. After 2–3 h, 20–27 mL of 70% nitric acid was added to the tubes while in water bath. After 2–3 h, deionized water was used to rinse the excess frustules stuck to slides into tubes. Falcon tubes were then placed in LEC Model K centrifuge at 2550 rpms for 8 min. After tubes were centrifuged, excess water was siphoned off and replaced with deionized water. This centrifuge and decanting process was repeated 6–8 times or until suspension was of a neutral pH. This process was developed through personal communication with diatomist and senior scientist, Dr. Mark Edlund, of the Science Museum of Minnesota [22].

2.3.2. Enumeration

Before processing, diatometer slides were viewed with light microscopy at 400× to verify that live diatoms dominated slide colonization. After processing, algal enumeration was conducted on three permanent slides from the left diatometer, three from the right diatometer, one from each of the triplicate composite samples, and two archived slides from past ANSP bioassessments of UTRC. Slides from past ANSP studies on UTRC were dated for May 1956, which coincides with the spring season of this GCSU study's sampling period. Algal enumeration was conducted by identifying and counting between 400 and 415 units, or valves (400 and 415 cells), with at least 60% of the valve intact along a transect. Although EPA periphyton standard protocol recommends counting to 600 valves, due to the high biodiversity found at this site, we counted only to at least 400. Valves were identified under oil immersion at 1000× on a Leica CTR5000 equipped with DIC and Leica DFC450C digital camera. After algal enumeration was conducted, raw data of triplicates and past slides were combined for each method (2018 composite sampling, 1956 composite sampling, left diatometer, and right diatometer). After triplicate data were combined, each treatment under separated taxonomic identification was assessed for and followed the 10 valves of 10 species rule according to EPA periphyton protocols [21]. We used Adobe Photoshop® to create diatom voucher flora Plates S1–S6 (Supplementary Materials) documenting the species found at this site for each method and morphological variability within species complexes: *Gomphonema parvulum sensu lato*, *Eunotia incisa sensu lato*, and *Tabellaria flocculosa sensu lato* [22].

2.3.3. Data and Statistical Analysis

To assess biodiversity and compare similarity at this UTRC site and between methods (diatometer and composite), the following were calculated: species richness, Shannon–Wiener diversity indices, Pielou's evenness, species relative abundances, Jaccard similarity indices and distances, and Sorenson index [22–25].

Analyses and dendrographs were conducted in R: A language and environment for statistical programming, with Vegan: Community Ecology Package, version 2.5–6 [26,27]. Jaccard dendrographs were generated using complete cluster analyses. Box and whisker plots comparing species richness across treatments, standard error, and confidence intervals calculations were generated in Systat 13 [28]. For ecological information, the literature was compiled for species or operational taxonomic units (OTUs) found to make up >1% relative abundance (RA) for our study. Percent relative abundance for UTRC for 2018 was calculated for separated taxa by combining all composite and diatometer triplicate observations. Percent relative abundance for UTRC for 1956 was calculated for separated taxa by combining observations from 1956 ANSP slides.

## 3. Results

### 3.1. Physicochemical Results

Physicochemical parameters were taken during this study's sampling period (Table 1). These readings show that our site was acidic and high in dissolved oxygen during our sampling period. The depths for these readings were recorded at 1.4 and 1.8 m. Means for all readings were as follows: temperature (17.68 ± 4.29 °C), DO % (86.83 ± 23.63), DO mg/L (8.44 ± 2.84), pH (5.12 ± 1.01), and conductivity mS/cm (10.75 ± 7.22).

**Table 1.** Physicochemical data for 2018 Upper Three Runs Creek study site at the time of deployment and retrieval of diatometers. Composite samples taken during diatometer retrieval.

| Sampling Date | DO (%) | DO (mg/L) | pH | Conductivity (mS/cm) | Depth (cm) | Temperature (°C) |
|---|---|---|---|---|---|---|
| 23 March deployment | 106.4 * | 11.09 | 6.46 | 13 | 140 | 13.5 |
| 9 April retrieval | 99.1 | 10.12 | 5.28 | 0.019 * | 177 | 14.45 |
| 15 August deployment | 88.7 | 7.85 | 4.17 | 15 | 177 | 21.33 |
| 30 August retrieval | 53.1 | 4.69 | 4.55 | 15 | 177 | 21.43 |

Note(s): * Denotes unreliable readings around sonde servicing.

### 3.2. Biodiversity

A total number of 4464 valves (cells) were counted for this study (817 across the 1956 samples and 3647 across the 2018 samples) belonging to a total number of 297 species/operational taxonomic units (OTUs). When combining taxa designations (i.e., grouping OTUs that are part of *G. parvulum*, *E. incisa*, and *T. flocculosa sensu lato* species complexes together), we found 259 species/OTUs. Voucher flora for this study documenting separated taxa and OTUs are found on Plates S1–S6 (Supplementary Materials). We found that for our separated taxa designations, of the 3647 counts, there were a total 218 species found in our 2018 samples (*n* = 9), and of the 817 counts in the 1956 samples (*n* = 2), there was a total of 155 species found. For taxa separated from *sensu lato* species complexes, we found that the 1956 composite sample had the highest species richness (*n* = 153), followed by the left diatometer (*n* = 129), 2018 composite sample (*n* = 111) and right diatometer sample (*n* = 67). Taxa combined or "lumped" together under past taxonomy followed similar trends as species richness findings (Tables 2 and 3). Overall, for separated taxa, treatments showed high biodiversity with Shannon–Wiener diversity indices ranging from (3.242 to 4.342) with the 1956 composite sample exhibiting the highest diversity and the 2018 samples exhibiting the lowest, however, still high. Pielou's evenness followed the same trends as Shannon–Wiener diversity exhibiting fairly high-to-high evenness with ranges 0.688–0.863. For separated taxa, species richness means for the 1956 composite samples (mean +/− sd = 99.5 ± 7.78) were greater than those for the left (75.33 ± 4.73) and right (43.67 ± 2.08) diatometers and 2018 composite samples (52.67 ± 19.86), and this

difference was significant (ANOVA, df = 3, *p* = 0.004). See Figure 2 for box and whiskers plots of species richness data by collection method type.

**Table 2.** Biodiversity and species richness for 2018 and 1956 samples (*Gomphonema*, *Eunotia*, and *Tabellaria* taxa separated from *sensu lato* species complexes), where d indicates diatometer sample and c indicates composite sample.

| Sample | Species Richness | SWDI (H′) | Pielou's Evenness (J′) |
|--------|------------------|-----------|------------------------|
| 2018 c | 111 | 3.242 | 0.688 |
| Right d | 67 | 3.263 | 0.776 |
| Left d | 129 | 3.661 | 0.753 |
| 1956 c | 153 | 4.342 | 0.863 |

**Table 3.** Biodiversity and species richness for 2018 and 1956 samples (*Gomphonema*, *Eunotia*, and *Tabellaria* taxa combined to *sensu lato* species complexes), where d indicates diatometer sample and c indicates composite sample.

| Sample | Species Richness | SWDI (H′) | Pielou's Evenness (J′) |
|--------|------------------|-----------|------------------------|
| 2018 c | 89 | 2.320 | 0.517 |
| Right d | 42 | 2.217 | 0.593 |
| Left d | 103 | 3.138 | 0.677 |
| 1956 c | 136 | 4.09 | 0.833 |

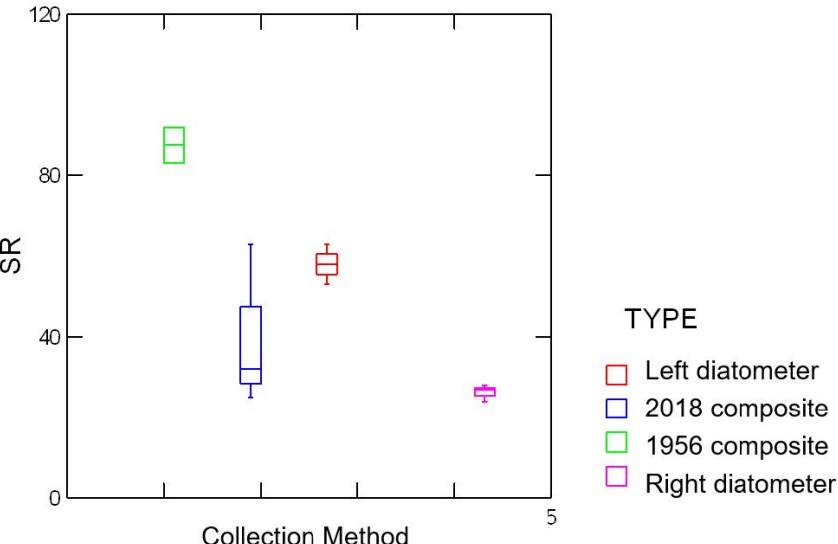

**Figure 2.** Box plots of species richness (SR) by collection method using data for separated taxa unit counts for *Gomphonema*, *Eunotia*, and *Tabellaria* species complexes.

*3.3. Similarity*

Jaccard and Sorenson indices for separated taxa followed similar trends with the shortest distance (greatest similarity) found between the right and left diatometers (0.658, 0.51), respectively, and the least similarity found between the 1956 composite samples and the right diatometer samples (0.817, 0.309), respectively. Samples analyzed by separating taxa clustered by Jaccard similarity reflected indices found in Table 4 and Figure 3 for right and left diatometers intersecting at a distance of (0.658). However, using the complete linkage cluster analysis, the right and left diatometer samples were found equally distant from the new composite samples. This hierarchical cluster analysis also showed the 2018 samples more similar to each other than to the 1956 samples. These findings differ from indices calculated from individual treatment pairs in Table 4.

**Table 4.** Jaccard and Sorenson similarity indices for 2018 and 1956 samples (*Gomphonema*, *Eunotia*, and *Tabellaria* taxa separated from *sensu lato* species complexes), where d indicates diatometer sample and c indicates composite sample.

| Sample Pair | Jaccard Similarity | Jaccard Distance | Sorenson |
|:---:|:---:|:---:|:---:|
| 2018 c & Right d | 0.219 | 0.781 | 0.360 |
| 2018 c & Left d | 0.218 | 0.782 | 0.358 |
| Right d & Left d | 0.347 | 0.658 | 0.510 |
| 2018 c & 1956 c | 0.211 | 0.789 | 0.348 |
| 1956 c & Left d | 0.236 | 0.763 | 0.383 |

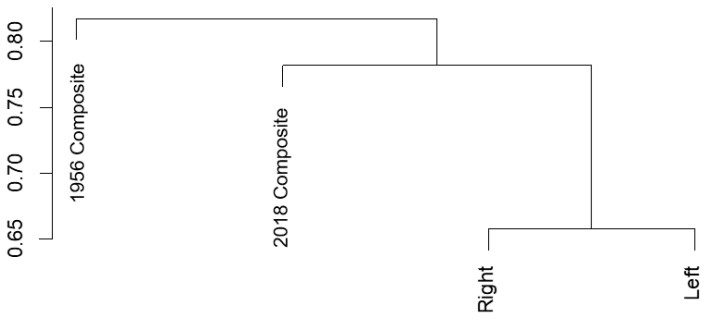

**Figure 3.** Jaccard dendrograph clustering collection methods (2018 composite, 1956 composite, right diatometer, and left diatometer) with separated taxa by Jaccard distance using complete linkage cluster analysis.

*3.4. Relative Abundance and Biological Indication*

*Gomphonema parvulum sensu lato*, *G. parvulum* Morphotype 2, and *G. parvulum* "protracted off-center" were found to be >1% relative abundance for the 2018 and 1956 samples (Table S2; Supplementary Materials). In this study, we found 10 taxa that were most likely combined into *G. parvulum sensu lato* in past studies, three of which have been raised to species level (*G. exilissimum* (Grunow) Lange-Bertalot and E. Reichardt 1996, *Gomphonema lagenula* Kützing 1844, and *Gomphonema confusum* Levkov, Mitic-Kopanja and E. Reichardt 2016), two that are recognized as a morphotypes (*G. parvulum* Morphotype 2 and *G. parvulum* Morphotype 3), and five separated in our analysis after documenting delineations (Plate S6: Figs.1–48, Plate S1: Fig. 2, and Table S1; Supplementary Materials).

Other taxa that we found to be >1% relative abundance include *Eunotia incisa* W. Smith ex W. Gregory 1854, *Eunotia rhomboidea* Hustedt 1950, *Eunotia* spp. girdle, *Fragilaria* Lyngbye 1819 spp. girdle, *Fragilariforma* D.M. Williams and Round 1988 spp. girdle, *Fragilariforma virescens* var. 1, *Frustulia crassinervia* (Brébisson ex W. Smith) Lange-Bertalot and Krammer 1996, *Frustulia saxonica* Rabenhorst 1853, *Gomphonema* spp. girdle, *Gomphonema preliciae* Levkov, Mitic-Kopanja and E. Reichardt 2016, *Luticola* D.G. Mann 1990 spp. girdle, *L. goeppertiana*, *Navicula leptostriata* Jørgensen 1948, *Navicula notha* J.H. Wallace 1960, Past "*Navicula minima* group", *Nitzschia* Hassall 1845 nom. cons. spp. girdle, *Nitzschia recta* Hantzsch ex Rabenhorst 1862, *Pinnularia* Ehrenberg 1843, nom. et typ. cons. spp. girdle, *Tabellaria flocculosa* "intermediate", and *Tabellaria* spp. girdle (Table S2; Supplementary Materials).

Dominant taxa only found in the past 1956 samples include *F. saxonica* and *G. preliciae*. *N. notha* were only found in composite samples (2018 and 1956). Taxa only found in the 2018 samples include *G. parvulum* "protracted off-center" and *F. virescens* var. 1. Taxa or OTUs that were found abundant in both the 2018 and 1956 samples had a higher relative abundance for the 2018 samples, except for *Gomphonema* spp. girdle. The OTU with the highest relative abundance in both the 2018 and 1956 samples was *Eunotia* spp. girdle with 31.5% for 2018 and 10% for 1956. The percent relative abundance calculated for the combined taxa was higher and, in most cases, over double the percentage for separated values: *E. incisa* (2018: 4.8%, 1956: 2.3%), *G. parvulum* (2018: 16.2%, 1956: 3.4%), and *T. flocculosa* "intermediate" (2018: 2.6%, 1956: n/a). In all cases, relative abundance was generally low. Of the 24 taxa, or OTUs, that were >1% relative abundance for the 2018 and 1956 samples, the literature reporting their ecological indication was found for 23 (Table S2; Supplementary Materials). Overall, the dominant species found indicate an acidic environment that is highly oxygenated and high in nutrients. These data are supported by physicochemical data that were collected at our study site (Table 1).

## 4. Discussion

For this UTRC site, separated taxa biodiversity and evenness indexes were high, scoring above 3 on a range of 0–5 and above 0.6 on a 0–1 range, respectively. These results were expected since this study site is part of a headwater stream located in a protected area with minimal land use and agricultural impacts. Composite values from 1956 were higher than the 2018 composite values, reflecting a possible decrease in biodiversity over time. However, only two slides for the 1956 composite samples were analyzed. The trend in decreasing diatom diversity over time has been noted in past ANSP reports [10]. Taxa combined or "lumped" together under past taxonomy followed similar trends. Although only two slides representing the composite sampling from 1956 were analyzed (due to availability and accessibility and season of sampling), it remains doubtful that an analysis of a third triplicate would yield results with a different trend; data found in the lower quartile for composite samples are still relatively higher than observations found in 2018 sampling methods. Results from an ANOVA test show that there is a significant difference across sampling methods. However, this may be due to the higher biodiversity present in 1956 or the variability found in new composite samples. Worth noting, the 1956 composite sampling site was most likely much further downstream from this study, according to ANSP report, which could account for differences between the 2018 and 1956 samples [29]. We did not sample here due to accessibility issues and because the exact location of the 1956 site remains unclear.

When comparing all sampling methods, the left diatometer species richness and diversity parameter indices were higher than other 2018 methods, both for combined and separated treatments. This result is interesting because left diatometer slides assessed for live diatom colonization prior to diatom enumeration were more heavily covered with diatoms. Right diatometer samples exhibited lower species richness and evenness indices. When comparing relative abundance indices from these samples, although right diatometer species richness is much lower than that of other 2018 samples, more species with higher abundances were found on them. Therefore, right diatometer slides had a higher biomass, but they did not exhibit higher diversity. Differences between diatometer methods could be attributed to differences in the amount of light exposure between banks, changes in stream flow or discharge, or depth. During field observations, it was noted that a small deep pool was located slightly downstream of the left diatometer [22]. Surrounding vegetation could also play a role in the difference found between diatometer methods. Vegetation on or near the left bank was different than that of the right bank, with tall pine and cypress trees on the left bank and grasses and smaller trees or shrubs on the right bank.

In assessing similarity for separated taxa, the right and left diatometers were found the most similar sharing about half to over half of their species in common. The 1956 composite samples and right diatometers were the least similar with only a 31% overlap. Given per-

centages of species overlap, we conclude that using all sampling methods is important, and one should not be chosen over the other when assessing species richness or investigating microbial biogeography and especially pertaining to environmental policy initiatives. These differences in percentage of shared species can be explained by biodiversity and evenness indices for these sample types. Because the dendrograph does not strictly match table values for pairs that were not the most similar, it is concluded that these differences could be largely due to the closeness of the values for similarity indices and Jaccard distances in these cases, and the binary structure of data for these tests and the complete linkage clustering method that was used. The focus and interest in this study is to investigate differences at the species level; therefore, the similarity metrics (Jaccard and Sorenson methods) used calculated values based on presence–absence and not abundances and after finding the maximum mathematical distances between points of two clusters. Because the right and left bank diatometers are almost the same "distance" from 2018 composite samples, they appear more similar to the 2018 composite sample than the 1956 composite samples. Here, the software chose the sample method with the greatest distance before combining them and creating a new matrix.

In analyzing relative abundances for 2018, the most abundant taxa belonged to the genus *Eunotia*, more specifically *Eunotia* girdle views. *G. parvulum* was also dominant in the 2018 samples (7.2% separated vs. a combined 16.2%). Because percentages were also found to increase by over double, the importance of splitting taxa in *G. parvulum* species complexes for future studies remains pertinent. In Europe, about 180 *Gomphonema* taxa are known and documented. In the southeastern U.S., the total of *Gomphonema* taxa are still being estimated. As recent studies have pointed out, continued study of this genus remains relevant as it is known to produce Janus cells, exhibit high variability, semi-cryptic taxa, and biogeography [16,30–34]. More species complexes were encountered in this study, including members of the *Tabellaria* and *Eunotia* genera. Because members of these genera and the *Gomphonema* genus account for 79.8% of the 2018 total counts, it is critical to distinguish morphotypes from species within these species complexes to gain better insight about diatom biodiversity at this site.

In our study, as shown in voucher flora Plates S1, S6, and Table S1 (Supplementary Materials), 10 taxa were found that were most likely combined in past assessments into the *G. parvulum* species complex. We intended to incorporate separations described in [30] and [33]. However, upon further inspection, we found the uniformity and delineation of the population micrographs and corresponding descriptions unclear from a taxonomic perspective. These studies also differed in separations and conclusions about results surrounding taxa, such as *G. exilissimum* and *G. lagenula*. However, both molecular studies either highlight the importance of continued morphotype identification or acknowledge the contribution of taxonomic varieties in water quality indication. Therefore, we chose a more conservative approach and separated *Gomphonema parvulum sensu lato* representatives based on observed differences within the population found in our study. For the purposes of water policy and understanding habitat water quality, it is recommended to split taxa and document morphotypes, OTUs, or ecotypes. Because diatoms reproduce asexually and rely on available conditions from their surrounding environment for cell wall production, documenting species morphotypes or ecotypes could answer questions about water quality that molecular studies may not.

In 1956, the ANSP reported a total of 84 taxa at UTRC. In this study, the diatom enumeration of the 1956 composite slides, 153 taxa (split), and 136 (lumped) are reported. However, species per slide and treatment (split or lumped) range from 83 to 105. For the 1956 UTRC studies, the ANSP found *Navicula mutica* Kützing 1844 to be the dominant taxon accounting for 15% relative abundance at the site [29]. In our study, the taxon identified to species level that was the most dominant for the 2018 samples was *E. rhomboidea* accounting for 8.3% for separated taxa methods. For combined taxa methods, *G. parvulum sensu lato* (16.2%) was the most abundant taxon identified to species level for 2018 samples. We found *L. goeppertiana* (5.4%) to be the most dominant taxon identified to species level in the

1956 samples. This result differs from that of the ANSP 1956 report. However, we were unable to determine if standard periphyton protocols were used in 1956 when collecting composite samples or during diatom enumeration. Information outlined in the reports documents the use of forceps, spoons, knives, and 17 mL vials, but whether or not habitats were sampled based on the percentage of site makeup is unclear. It also remains unclear how many valves were counted per sample; although, it is suspected to be in the thousands. In the 1956 report, the ANSP documents that members of the genera *Eunotia*, *Tabellaria*, *Pinnularia*, and *Achnanthes* Bory 1822 were common in all samples. With the exception of *Achnanthes* representatives, findings from our study are the same as those of the ANSP in 1956 with respect to the common genera found.

According to dominant diatom species ecological indication, UTRC is a nutrient rich and acidic environment, and according to 2018 physicochemical data, UTRC is an acidic environment with high dissolved oxygen levels (Supplementary Materials). These findings are the same as those in 1956, as the ANSP reports that dominant genera represent "cool, slightly acidic streams" and indicate that the "river has a high nutrient level" [29]. *G. parvulum sensu stricto* is frequently reported from eutrophic waters, while *G. exilissimum* is found in oligotrophic habitats. Both species are documented here at UTRC, which is considered a reference site that has been protected for at least 50 years. Therefore, nutrient levels reflected by the bioindicators found in this study are more likely contributed by a natural source (e.g., vegetation falling into the system) than land use or agriculture. However, continued evaluation of the *G. parvulum* complex is necessary to understand the ecological preferences of representatives.

During field observations, it was noted that UTRC is high in tannins, which could explain the acidity found at this site. Physical aspects of this site include a high sediment load with a mostly sandy bottom and some grass-like and woody vegetation. Sandy bottom creeks are typical for the coastal plain, which plays a role in driving algal communities. Other references that document coastal plains diatoms include the [35] Gaiser and Johansen 2000 study. Some of the dominant taxa in their study were also found in this 2018 study.

In summary, separated taxa biodiversity and evenness indexes were high for this site with a trend of a possible decrease in biodiversity over time. For this study, only two slides were available for analysis. When comparing past archives in future studies, we recommend using more archives where available. Results also demonstrated a significant difference in species richness across sampling methods, and the right and left diatometers were found to only share about half to over half of the same species. Therefore, we conclude that using all sampling methods (composite, right bank, and left bank diatometers) is important for conducting future diatom bioassessments.

For a clearer understanding of environmental indication, we recommend finely separating taxa within species complexes (e.g., Supplementary Materials Plates S6 and Table S1; [13,16,36,37]) and measuring nutrient levels for future studies on diatoms as biological indicators. The most abundant taxa in the 2018 samples belonged to the genera *Eunotia* (*Eunotia* girdle views and *E. rhomboidea*) and *Gomphonema* (*G. parvulum*). These representatives of these genera are found in acidic environments (Supplementary Materials Table S2; [38–43]). The *Eunotia* taxa in this study are found in oligotrophic waters, while *G. parvulum* is typical of eutrophic environments, but has also been observed in oligotrophic waters (Supplementary Materials Table S2; [16,38–46]). These genera are known to comprise species complexes, which could contain species that indicate something different about water quality. For example, *G. exilissimum* was once included in *G. parvulum sensu lato* species complex but has since been elevated to species level. *G. parvulum* is known to indicate high nutrients, while *G. exilissimum* is typically found in oligotrophic waters [47]. Both taxa were found in this study. Separating taxa and documenting morphotypes, ecotypes, and OTUs as recommended could also aid in gleaning information about the surrounding aquatic environment that would not be observed in molecular studies (e.g., valve morphogenesis and diatom teratologies) [48].

**Supplementary Materials:** The following supporting information can be downloaded at: https://www.mdpi.com/article/10.3390/w15142578/s1. Plates S1–S6: Diatoms at Upper Three Runs Creek site in 2018. Table S1. Morphological features distinguishing taxa originally belonging to *Gomphonema parvulum sensu lato* species complex for 2018 Upper Three Runs Creek study. Table S2. List of diatom taxa and operational taxonomic units (OTUs) found at >1% relative (RA) identified from 2018 samples and 1956 archived slides collected from Upper Three Runs Creek, South Carolina.

**Author Contributions:** Conceptualization, K.M.J. and K.M.M.; methodology, K.M.J.; software, K.M.J.; validation, K.M.M. and E.G.; formal analysis, K.M.J.; investigation, K.M.J.; resources, K.M.M.; data curation, K.M.J. and K.M.M.; writing—original draft preparation, K.M.J.; writing—review and editing, K.M.J., K.M.M., and E.G.; visualization, K.M.J.; supervision, K.M.M.; project administration, K.M.M.; funding acquisition, K.M.M. All authors have read and agreed to the published version of the manuscript.

**Funding:** This research was funded by Watershed Protection Branch, Environmental Protection Division Georgia Department of Natural Resources, grant number 58 6002054 to K.M.

**Data Availability Statement:** Not applicable.

**Acknowledgments:** Many thanks to the editor and two reviewers for their valuable suggestions for our manuscript. We thank Iowa Lakeside Laboratory, the Friends of Lakeside and the Kingston Fellowship at Lakeside Laboratory, University of Iowa, Mark Edlund, UGA-SREL, and everyone in the Manoylov Phycology lab, Georgia College & State University, and the George M. Barley, Jr., Endowment at the Institute of Environment, Florida International University. We are grateful to Marina Potapova and Laura Aycock, for providing diatom slides from the diatom collection of the Academy of Natural Sciences, Drexel University and Rosalina Hristova for her aid in research. This work was part of the first author's Master's graduate research at the Department of Biological and Environmental Sciences and the Graduate School at Georgia College & State University.

**Conflicts of Interest:** The authors declare no conflict of interest.

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
