# Peer review of "Diatom Voucher Flora and Comparison of Collection and Taxonomic Methods for Biodiversity Hotspot Upper Three Runs Creek"

_water, doi:10.3390/w15142578_

Round 1
Reviewer 1 Report
Lines 99-101: What does Europe have to do with North America? This statement does not make sense to me. I recommend you reorganize the sentence to lead off with the importance of the work in Europe.
Line 238: Add a period between "1.8 m" and "Means"
I think there should be more references that support your conclusions. The discussion does not support the last statement of the abstract, that this is an oligotrophic environment. In fact, it is stated that the common diatoms are consistent with an eutrophic environment. You need stronger literature support for that ending statement of the abstract.
It would be good to include a conclusion section that reiterates your findings and summarizes the discussion.
Author Response
Dear Editor and Reviewers,
Thank you for the time you have spent reviewing our manuscript. We appreciate the time you have taken, and the consideration and suggestions you have made. Below we have addressed each edit and suggestion. Please see the manuscript for highlighted portions of any changes. I did not see where I could access the Appendix on the portal to change the citation numbering to agree with the additions, so I numbered the new citations at the end so not to interfere with the numbering in the Appendix. Any identical text is from my web available thesis, which has been cited in this manuscript as well. We look forward to the next round of edits. Please let us know if you have any questions.
Thank you, Katherine
Reviewer 1
Thank you for your valuable suggestions.
Are the conclusions supported by the results? Our conclusions were strengthened |
Lines 99-101: What does Europe have to do with North America? This statement does not make sense to me. I recommend you reorganize the sentence to lead off with the importance of the work in Europe.
We clarified our statement:
The majority of past phycological work has been conducted in Europe and American diatomists have used European names and metrics to infer regional and national water quality. This has led to assigning European published names to southeastern diatom species, which could change perspectives on freshwater conditions in these areas. Because algal research conducted throughout Europe remains a primary literature source for diatom identification, documentation of North American diatom species and their autecology is imperative [14-16].
Line 238: Add a period between "1.8 m" and "Means"
Done: 1.8 m. Means
I think there should be more references that support your conclusions. The discussion does not support the last statement of the abstract, that this is an oligotrophic environment. In fact, it is stated that the common diatoms are consistent with an eutrophic environment. You need stronger literature support for that ending statement of the abstract.
Highlighted and changed to (in abstract lines 29-30): Diatom community composition indicated an acidic and highly oxygenated environment. In table 2 in Appendix, we show the extensive literature reporting taxa from variable environments. We report findings in a habitat that has not been altered in terms of anthropogenic alterations due to nutrient additions in 50 years.
It would be good to include a conclusion section that reiterates your findings and summarizes the discussion.
Included:
In summary, separated taxa biodiversity and evenness indexes were high for this site with a trend of a possible decrease in biodiversity over time. For this study only two slides were available for analysis. When comparing past archives in future studies, we recommend using more archives where available. Results also demonstrated a significant difference in species richness across sampling methods and right and left diatometers were found to only share about half to over half of the same species. Therefore, we conclude that using all sampling methods (composite, right and left bank diatometers) is important for conducting future diatom bioassessments.
For a clearer understanding of environmental indication, we recommend separating taxa within species complexes and measuring nutrient levels for future studies on diatoms as biological indicators. The most abundant taxa in 2018 samples belonged to the genera Eunotia (Eunotia girdle views and E. rhomboidea) and Gomphonema (G. parvulum). These representatives of these genera are found in acidic environments (Appendix B; [38-43]). The Eunotia taxa in this study are found in oligotrophic waters, while G. parvulum is typical of eutrophic environments, but has also been observed in oligotrophic waters (Appendix B; [16, 40, 42, 44]). These genera are known to comprise species complexes, which could contain species that indicate something different about water quality. For example, G. exilissimum was once included in G. parvulum sensu lato species complex but has since been elevated to species level. G. parvulum is known to indicate high nutrients, while G. exilissimum is typically found in oligotrophic waters [47]. Both taxa were found in this study. Separating taxa and documenting morphotypes, ecotypes, and OTUs as recommended could also aid in gleaning information about the surrounding aquatic environment that would not be observed in molecular studies (e.g., valve morphogenesis and diatom teratologies) [48].

Reviewer 2 Report
Johnson et al. provide an important analysis and re-investigation of the diatoms of Upper Three Runs Creek, a tributary of the Savannah River, and a site that has importance in the history of environmental monitoring. The authors provide appropriate information about the sampling history of this site, and the challenges of working with historical reports, data, and samples, and comparing them to more recently collected samples. Their approach is clearly explained, as is their rationale for doing this study. Their methodology was appropriate, and they did a good job of clarifying factors that affected their sampling, and their results. Their taxonomic conclusions and comparisons are reasonable, and well-explained. Their voucher information will be important for better documentation and understanding of the diatom flora of SE rivers and streams. The main suggestion I have would be to include more details at the end of the discussion about the usefulness and importance of doing these types of studies, and recommendations on how these types of studies can be used. The manuscript seemed to end abruptly.
Generally, the english is good. There were a few minor places where the words should have been in the plural form, and I'd suggest the authors read through the manuscript an additional time to catch those places.
Author Response
Dear Editor and Reviewers,
Thank you for the time you have spent reviewing our manuscript. We appreciate the time you have taken, and the consideration and suggestions you have made. Below we have addressed each edit and suggestion. Please see the manuscript for highlighted portions of any changes. I did not see where I could access the Appendix on the portal to change the citation numbering to agree with the additions, so I numbered the new citations at the end so not to interfere with the numbering in the Appendix. Any identical text is from my web available thesis, which has been cited in this manuscript as well. We look forward to the next round of edits. Please let us know if you have any questions.
Thank you, Katherine
Reviewer 2
The main suggestion I have would be to include more details at the end of the discussion about the usefulness and importance of doing these types of studies, and recommendations on how these types of studies can be used. The manuscript seemed to end abruptly.
Same as above. Highlighted and added lines 450-473
Generally, the english is good. There were a few minor places where the words should have been in the plural form, and I'd suggest the authors read through the manuscript an additional time to catch those places.
Read and highlighted changes
Overall
In particular, the authors should extend the reference list in order to better sustain the statements done in the Abstract and Conclusions, as well as to include more details at the end of the discussion about the importance of doing these types of studies.
Same as above. Highlighted and added lines 450-473. Added references found in appendix to text for better support
Notes: found some typing errors in Appendix, typing errors to be corrected in appendix Table: Circumneutral spell corrected and “Ground water” to Groundwater.
